A diverse global fungal library for drug discovery

Niu Guodong 1
Annamalai Thirunavukkarasu 2
Wang Xiaohong 1
Li Sheng 3
Munga Stephen 4
Niu Guomin 5
Tse-Dinh Yuk-Ching 2 6
Li Jun lij@fiu.edu 1 6
1 Department of Biological Sciences, Florida International University , Miami , FL , United States of America
2 Department of Chemistry and Biochemistry, Florida International University , Miami , FL , United States of America
3 School of Public Health, City University of New York , NY , United States of America
4 Center for Global Health Research, Kenya Medical Research Institute , Kisumu , Kenya
5 Department of Hematology, Southern Medical University Affiliated Nanhai Hospital , Foshan , Guangdong , China
6 Biomolecular Sciences Institute, Florida International University , Miami , FL , United States of America
Gomez Shawn
Electronic publication date: 2020 Nov 27
Publication date: 2020
Volume: 8
Electronic Location ID: e10392
Received 2020 Jun 25; Accepted 2020 Oct 28
Copyright: ©2020 Niu et al.
Copyright year: 2020
Copyright holder: Niu et al.
License: This is an open access article distributed under the terms of the Creative Commons Attribution License, which permits unrestricted use, distribution, reproduction and adaptation in any medium and for any purpose provided that it is properly attributed. For attribution, the original author(s), title, publication source (PeerJ) and either DOI or URL of the article must be cited.
License URL: https://creativecommons.org/licenses/by/4.0/

Keywords: Fungus, Secondary metabolites, Small molecule, Natural product, Drugs, Anti-malaria, Antibiotics, Anti-leukemia

Funding: NIAID No. 1R01AI125657 NSF Career Award No. 1453287 This work is supported by NIAID (No. 1R01AI125657) and NSF Career Award (No. 1453287). The funders had no role in study design, data collection and analysis, decision to publish, or preparation of the manuscript.

==============================
Background

Secondary fungal metabolites are important sources for new drugs against infectious diseases and cancers.

Methods

To obtain a library with enough diversity, we collected about 2,395 soil samples and 2,324 plant samples from 36 regions in Africa, Asia, and North America. The collection areas covered various climate zones in the world. We examined the usability of the global fungal extract library (GFEL) against parasitic malaria transmission, Gram-positive and negative bacterial pathogens, and leukemia cells.

Results

Nearly ten thousand fungal strains were isolated. Sequences of nuclear ribosomal internal transcribed spacer (ITS) from 40 randomly selected strains showed that over 80% were unique. Screening GFEL, we found that the fungal extract from Penicillium thomii was able to block Plasmodium falciparum transmission to Anopheles gambiae, and the fungal extract from Tolypocladium album was able to kill myelogenous leukemia cell line K562. We also identified a set of candidate fungal extracts against bacterial pathogens.

Introduction

Natural products, produced by living organisms in nature, have been used as medicine for thousands of years (Dias, Urban & Roessner, 2012; Buyel, 2018). For instance, the treatment of malaria was recorded in China with the Qinghao plant (Artemisia annua) as early as the second century. The active ingredient, qinghaosu (artemisinin), was isolated from the plant by Youyou Tu and her colleagues in 1971 (Luo & Shen, 1987; Hsu, 2006). The establishment of microbiology in the early modern era led to drug discoveries from microbes. The first antibiotic, penicillin, was discovered from a fungus by Alexander Fleming (Fleming, 1929). Fungi have initially been and still are used to produce medicines to treat infectious diseases (Elder, 1944). Furthermore, people use natural products to treat tumors. Indeed, more than half of anti-tumor drugs or leads in current clinical trials are from natural products (Wolfender & Queiroz, 2012).

Microbial metabolites are essential resources for drug discovery (Lenzi et al., 2018). Compared with other natural product resources, fungi have the following advantages. First, there are enormous fungal species: about 120,000 fungal species have been described (Hawksworth & Lucking, 2017) and 5.1 million fungal species are estimated (Blackwell, 2011). Second, fungi produce broad and diverse secondary metabolites with a vast difference in chemical structures (Pham et al., 2019). Third, large-scale fermentation can generate a large amount of fungal secondary metabolites, which was exampled by the production of alcohol and lactic acid. However, yield of many target fungal secondary metabolites is restricted by fungal growth and differentiation (Nielsen & Nielsen, 2017; Keller, 2019; Pham et al., 2019), which is resolved by new technologies that enable us to engineer a fungus to produce a specific compound in high yield by modifying its metabolic pathways (Van Dijk & Wang, 2016). Also, the recent development of genomic sequencing technology and the identification of more biosynthetic gene clusters accelerate the discovery and application of new compounds from fungi (Hussain et al., 2017; Keller, 2019).

Fungi can be isolated from soil, water, air, plants, or other organisms. In particular, the endophytic fungi from the plants can generate similar secondary metabolites as their hosts (Venieraki, Dimou & Katinakis, 2017). Thus, the heterologous expression can replace their hosts as the supplies of crude materials for some medicines (Van Dijk & Wang, 2016). Diverse fungal libraries are critical for the research and industry communities (Niu et al., 2015). At present, there are several specific fungal libraries for drug discovery (May et al., 2004; Richards et al., 2012; Zhang et al., 2014), many of which focus on specific environments (Li et al., 2005; Gonzalez-Menendez et al., 2018; Zhang et al., 2018). We focus on generating a global diverse fungal library to facilitate new drug discovery. The soil and plant samples were collected globally, currently including from Asia, Africa, and North America.

To determine the usability of our fungal library, we screened the newly established fungal extract library for malaria transmission inhibitors, antibiotics, and drug leads against chronic myeloid leukemia. A set of positives were discovered.

Materials & Methods

Collecting samples

We collected plant and soil samples from different regions around the world. The field collection was approved by the United States Department of Agriculture with permit number of P526P-18-03319. The soil samples were taken in 5–10 cm depths under the surface. Plant samples consisted of the whole plant or separated plant parts such as roots, stems, leaves, flowers, fruits, or various combinations of components. Samples were stored on ice or in a 4 °C fridge immediately after collection. No samples were more than 5 g. Most of samples collected by authors, residents, and friends were mailed to labs and processed locally. For instance, samples collected in China and Myanmar were shipped to Guomin Niu’s lab in Guangdong, China to process. Samples collected in the US were processed at Jun Li’s lab in Florida, USA. The fungal extracts form the GFEL library.

Isolation of fungi from soil and plants

For each soil sample, we transferred 50 mg soil to a 1.5 mL plastic tube, and one mL autoclaved distilled water was added. The sample was vortexed for 30 secs and centrifuged at 500 g for 2 min (min) to get rid of the soil particles. For plant samples, the first step was to sterilize the plant surface by rinsing the sample with distilled H2O, then soaking in 70% ethanol for 10 s (sec) and rinsing again with distilled H2O. After sterilization, the plants were cut into 0.5 cm × 0.5 cm pieces and transferred into a sterile mortar. Then, two mL of distilled H2O was added, followed by grinding with a pestle for 1–3 min and the slurry was transferred to a 1.5 mL plastic tube and then centrifuged at 500 g for 2 min to get rid of the particles. About 100 µL of upper supernatant from treated soil or plants was evenly spread onto a 100 × 15 mm Petri Dish plate containing 14 mL Malt Extract Agar (MEA), made of 10 g malt extract, 1 g yeast extract, 15 g agar, and 0.05 g chloramphenicol (Sigma-Aldrich, St. Louis, MO) in 1 L distilled H2O and autoclaved at 121 °C for 20 min. The plates were sealed with parafilm, and the fungi were allowed to grow for 7–14 days at room temperature (RT) with cycles of 12 h (hr) of darkness and 12 hr of light.

The fungal colonies on the MEA medium plates were picked with a toothpick and inoculated in a new MEA plate by streaking. If the colonies were mixtures of two or more species, we kept inoculating and streaking until a single colony appeared. Finally, a piece of fungal agar containing mycelium or spores was cut and transferred to a 1.5 mL Eppendorf tube containing 500 µL of sterile 20% glycerol in distilled H2O. We stored the cells in a −80 °C freezer for long-term storage.

Metabolite production and extraction

Cereal based medium was used to grow fungi to produce secondary metabolites (Niu et al., 2015). Briefly, six pieces of Cheerios Breakfast cereals (General Mills, Minneapolis, MN) were placed in a glass test tube, capped with a plastic lid and autoclaved for 20 min, and then two mL sterile sucrose water (3 g of sucrose and 50 mg chloramphenicol in 1 L distilled H2O) was added into the tube. Later, the fungal colony grown on the MEA plate was inoculated into the cereal medium and incubated at RT for one month to produce sufficient metabolites. A month late, two mL ethyl acetate was added into a tube to extract the fungal metabolites, mixed with a glass stirring rod, and placed overnight in a chemical hood with gentle shaking. The next day, one mL upper layer of supernatant was transferred to a 1.5 mL tube. After centrifugation (2,000 g for 2 min), around 950 µL clear supernatant was transferred to a pre-weighed 1.5 mL plastic tube and dried with SpeedVac concentrator (Thermo Fisher Scientific, Waltham, MA). Finally, the dry extracts were weighed and dissolved in an appropriate amount of dimethyl sulfoxide (DMSO) to prepare 10 mg/mL stock solution and stored in a −20 °C freezer for future screening assays.

Determination of fungal species

Forty fungal isolates were randomly picked to evaluate the fungal library’s diversity. The fungi were cultured with liquid malt extract medium at RT for one week, and mycelium was collected for DNA extraction using DNAzol (Thermo Fisher). Genomic DNA applied as PCR templates were isolated using DNAzol Reagent following the manual (Thermo Fisher Scientific). To identify the fungal species, nuclear ribosomal ITS regions were amplified by PCR with specific primers (Table 1) (Op De Beeck et al., 2014) using the following approach: 94 °C 2 min; 94 °C 30 s, 55 °C 30 s, 72 °C 1 min, 35 cycles; 72 °C 5 min. The amplified products were sequenced and blasted against the NCBI database to identify fungal species (Raja et al., 2017).

Table 1 PCR Primers for fungal ITS regions.

Primer name	sequences	rDNA region	
ITS1F (F)	CTTGGTCATTTAGAGGAAGTAA	18S	
ITS2 (R)	GCTGCGTTCTTCATCGATGC	5.8S	
ITS3 (F)	GCATCGATGAAGAACGCAGC	5.8S	
ITS4 (R)	TCCTCCGCTTATTGATATGC	28S	
ITS86F (F)	GTGAATCATCGAATCTTTGAA	5.8S	
ITS86R (R)	TTCAAAGATTCGATGATTCAC	5.8S	

Screening the fungal extract library to identify malaria transmission-blocking candidates with ELISA assays

As described previously (Niu et al., 2015), the red blood cells (iRBC) infected by Plasmodium falciparum (NF54 strain from MR4, Manassas, VA) were cultured in RPMI-1640 medium (Life Tech, Grand Island, NY) supplemented with 10% heat-inactivated (56 °C for 45 min) human AB+ serum (Interstate blood bank, Memphis, TN), 12.5 µg/mL hypoxanthine and 4% hematocrit (O+ human blood) in a candle jar at 37 °C for 15-17 days. The medium was replaced every day to provide sufficient nutrients. Blood smears stained with Giemsa (Sigma-Aldrich, St. Louis, MO) were used to examine parasitemia or gametocytemia every other day under a light microscope. Then, the cells were collected and washed three times with RPMI-1640 at 300 × g for 4 m. The cell pellets were re-suspended in PBST (PBS containing 0.2% Tween-20) and homogenized by ultra-sonication with six cycles of 10 s pulse and 50 s resting on ice for each period. The lysates were centrifuged at 8,000 g for 2 min to remove insoluble materials and cellular debris. With the iRBC lysate and insect cell-expressed recombinant FREP1, the ELISA assay was used to screen the fungal extract library to block FREP1-parasite interaction (Niu et al., 2015). A 96-well ELISA plate was coated with 50 µL iRBC lysate (2 mg/mL protein) and incubated overnight at 4 °C. After coating, the plate was blocked with 100 µL of PBS plus 0.2% bovine serum albumin (BSA) per well for 1.5 hr at RT. After removal of the blocking solution, FREP1 (10 µg/mL) in blocking buffer (PBS plus 0.2% BSA) was added to each well, and 1 µL fungal extract was taken from a 96-well plate containing 2 mg/mL crude extract dissolved in DMSO in each well with a multiple-channel pipette and transferred to the ELISA plate, then incubated for 1 hr at RT with gentle shaking. After washing three times with PBST, 50 µL rabbit anti-FREP1 polyclonal antibody (Niu et al., 2015) (diluted 1: 5,000 in blocking buffer, 1 µg/mL) was added to each well and incubated for 1 hr at RT. About 50 µL alkaline phosphatase-conjugated anti-rabbit IgG (diluted 1: 20,000 in blocking buffer) was added to each well and incubated for 45 min at RT. The wells were washed three times with PBST between incubations. After washing, each well was developed with 50 µL pNPP substrate (Sigma-Aldrich) until the colors were visible, and absorbance at 405 nm was measured. The functional FREP1 supplemented with 1 µL solvent (DMSO) was used as non-inhibition control, and the heat-inactivated FREP1 (65 °C for 15 min) was used as a 100% inhibition control.

Determination of the transmission-blocking activity of the fungal extracts in mosquitoes

Following the previous protocol (Zhang et al., 2015), the 15- to 17-day old cultured P. falciparum containing 2–3% stage V gametocytes were collected and diluted with new O+ type human blood to get 0.2% stage V gametocytes in the blood. Then, the 150 µL blood was mixed with the same volume of heat-inactivated AB+ human serum. Then, 3 µL candidate fungal extract in DMSO (10 mg/mL or 2 mg/mL) was mixed with 297 µL infected blood, the final fungal extract concentration in blood was 100 or 20 µg/mL, respectively. SMFA was performed to feed about 100 3–5 days old An. gambiae G3 female mosquitoes for 15 min, and the engorged mosquitoes were maintained with 8% sugar in a BSL-2 insectary (28 °C, 12-h light/dark cycle, 80% humidity). The midguts were dissected seven days post-infection and stained with 0.1% mercury dibromofluorescein disodium salt in PBS for 16 min. The oocysts in midguts were counted under a light microscope. The standard membrane feeding assays were conducted at least twice to confirm the results.

Screening the fungal extract library to identify antibiotics

A subset of fungal extracts, randomly selected, were tested for the antibacterial activity to inhibit the growth of Shigella flexneri (ATCC 9199), Staphylococcus aureus (ATCC 14775), methicillin-resistant Staphylococcus aureus (MRSA, ATCC BAA-44) and E. coli (AS17tolc). The drug-screening was performed in a 384-well microplate format using the following procedures. Bacteria were cultured in 50 mL Mueller Hinton broth (MHB) in a 150-mL flask overnight at 37 °C. The next day, the cells were first OD600 adjusted to 0.1 and then further diluted 1:100 in MHB, and a volume of 50 µL (∼105 CFU) is added to each well of the 384-well sterile microplates (Thermo Fisher Scientific). Fungal extracts (0.5 µL in DMSO) were then added to each test well at a final concentration of 40 µg/mL. The plates were incubated for 18–20 hr at 37 °C. At the end of this incubation, resazurin (Sigma-Aldrich) was added to the wells to determine the growth of bacteria. The final concentration was 0.02%, and the plates were further incubated for 4-6 hr at 37 °C. In the presence of viable cells, resazurin was reduced to resorufin (pink) along with an increase in fluorescence (O’Brien et al., 2000). Extracts showing antibiotic activity (hits) were scored as those that prevented the color change and also reduced the fluorescence (Ex 540, Em 590nm) by 90% when compared to the control wells containing no inhibitor. Ciprofloxacin was used as a positive control for bacterial growth inhibition. Three wells were used for each sample. For the positive candidate extracts, we repeated the experiments at least once to confirm the results.

Screening the fungal extract library to identify drugs leads against chronic myeloid leukemia with MTT assays

Cell proliferation was analyzed by the 3-(4,5-dimethylthiazol-2-yl)-2,5-diphenyl tetrazolium bromide (MTT) using Vybrant® MTT Cell Proliferation Assay (Thermo Fisher Scientific) with the human immortalized myelogenous leukemia cell line K562. Around 2 × 104 cells in 100 µL culture medium (RPMI 1640 + 2mM glutamine + 10% fetal bovine serum) were seeded in 96-well microplates and incubated at 37 °C with 5% CO2. The next day, one µL fungal extract in DMSO was added (final concentration of the fungal extract was 20 µg/mL), and the cells were incubated at 37 °C with 5% CO2 for another 24 hr. Next, the microplate was centrifuged at 500 g for 10 min to pellet the cells, the medium was carefully removed as much as possible, and 100 µL of fresh medium was then added. About 10 µL of the 12 mM MTT stock solution was added, mixed, and incubated for 4 hr at 37 °C. The microplate was centrifuged again at 500 g for 10 m. After removing 75 µL of the medium from the wells with 25 µL medium with cells left, 50 µL of DMSO was added to each well, mixed, and incubated at 37 °C for 10 min to dissolve formazan crystal for measurement. The same amount of DMSO without drugs was applied as a control. The optical density was measured at an absorbance wavelength of 540 nm. Cell growth inhibition rate (%) = (A540 of control −A540 of treatment/ A540 of control) ×100%. Triplicates were conducted for each sample.

Statistical analysis

All the experiments were independently repeated at least twice and analyzed with the Wilcoxon-Mann–Whitney test using GraphPad Prism (GraphPad Software, CA, USA).

Sequence availability

All fungal ITS sequences obtained in this project have been deposited into GenBank at NCBI (https://www.ncbi.nlm.nih.gov/genbank/sequenceids/).

Construction of phylogenetic tree

Forty fungal ITS sequences in FASTA format were input into an online multiple sequence alignment tool (https://www.ebi.ac.uk/Tools/msa/clustalo/) using Clustal Omega algorithm (Higgins & Sharp, 1988). The parameter “DNA” was selected for input sequence and “ClustalW” was selected as output format. All other parameters were kept as defaults. Multiple sequence alignment was conducted by clicking on “Submit”. After alignment was completed, the tab of “Phylogenetic Tree” was clicked and the checkbox “Real” was selected for “Branch length” to visualize the phylogenetic tree. The phylogenetic tree was saved as a pdf file through “print” under “File”.

Results and Discussion

Extensive fungal library with nearly ten thousand isolates

A large and diverse fungal library is powerful in discovering new drugs. To achieve this goal, we collected samples globally. Current collection includes samples from Kenya, Myanmar, USA, and China. We received about 2395 soil samples and 2324 plant samples. We collected the whole plant or separated plant parts such as roots, stems, leaves, flowers, fruits, or various combinations of components. The samples were from 36 regions, including Nairobi in Kenya, 10 regions in the United States, Yangon in Myanmar, and 24 districts in China (Table 2). The collection places cover various climate zones.

From these samples, 9,053 fungal isolates in total were cultured. Among them, 2,356 were from the plant samples, and 6,688 were from soil samples. About one fungal strain per plant-part sample and 2.8 fungal isolates per soil sample were obtained by average. Nearly 69.4% of fungal isolates were from the subtropical climate in China (Shanghai, Guangzhou, and Chongqing) and the USA (e.g., Dallas, New Orleans, and Oklahoma City). About 8% of fungal isolates were from tropical climates such as Miami in the USA, Yangon in Myanmar, Kisumu in Kenya. Approximate 4% were from tropical/subtropical highland climate areas such as Nairobi in Kenya, the Lijiang National Park, and the Potatso National Park in China. A small portion (2.1%) was from the cold areas, such as Jiuzhaigou National Park in China, and Alaska in the USA (Fig. 1A). We collected the samples from different landforms, including hills, mountains, plateaus, canyons, valleys, and bays. Notably, some fungi were isolated from samples collected from the mountains with an altitude over 3,000 m, such as the Cang Mountains and Meili Snow Mountains in the Yunnan Province, China. The vast difference of sample collection in location, weather, climate, and altitude (Fig. 1B) promises the diversity of fungal species and their genetic background.

To maximize the diversity and quantity in our fungal library, we selected fungal colonies based on their location, color, and morphology on culture plates. Therefore, the fungal isolates in the library look strikingly different (Fig. 1C). This visible criterion facilitates the fungal library construction. However, we also discarded many fungi that are different species with similar morphology. We examined the species diversity of our fungal library at the molecular level by randomly picking 40 fungal isolates. Their genomic DNA was isolated, the ITS regions were amplified, and PCR products were sequenced. These sequences have been deposited into GenBank at https://www.ncbi.nlm.nih.gov/genbank/sequenceids/. Their Accession numbers are MT594355–MT594393 and MT584204. We searched these sequences against NCBI DNA databases using blast. Results show that about 12.5% of total fungal isolates have identical ITS sequences to others in the library (Table 3). For instance, 126-G10, 117-B9, and 45-F10 have identical ITS to Fusarium solani, and 2 of 3 might be duplicates. Strains 99-H5 and 78-D10 have identical ITS to Penicillium sclerotiorum. More than 80% of fungal isolates belong to different species or strains, indicating the fungal library is highly diverse. A small portion (<12.5%) are duplications of the other. Based on these ITS sequences, a phylogenetic tree was constructed, displaying the fungal diversity in samples (File S1). Three genera (Trichoderma, Fusarium and Penicillium) present in three big branches, which is consistent to our sampling sources, e.g., soil and plants. Trichoderma and Fusarium are the most prevalent soil fungi and many are associated with plants (Harman et al., 2004). Penicillium is ubiquitous genus with more than 350 species already identified (Visagie et al., 2014).

Table 2 Fungal strains collected from different locations of the world.

# of isolates	Area/City/County	State/Province	Country	Continent	
188	Norman	Oklahoma	USA	North America	
287	Oklahoma City	Oklahoma	USA	North America	
208	Stillwater	Oklahoma	USA	North America	
76	Dallas	Texas	USA	North America	
68	New Orleans	Louisiana	USA	North America	
59	Pensacola	Florida	USA	North America	
47	Tallahassee	Florida	USA	North America	
33	Lake City	Florida	USA	North America	
401	Miami	Florida	USA	North America	
119	Juneau	Alaska	USA	North America	
702	Nairobi	NA	Kenya	Africa	
322	Yangon	NA	Myanmar	Asia	
181	Shanghai	NA	China	Asia	
34	Beijing	NA	China	Asia	
39	Chongqing	NA	China	Asia	
1469	Foshan	Guangdong	China	Asia	
472	Guangzhou	Guangdong	China	Asia	
1093	South China Botanical Garden	Guangdong	China	Asia	
667	Taishan	Guangdong	China	Asia	
37	Zhongshan	Guangdong	China	Asia	
45	Jiangmen	Guangdong	China	Asia	
14	Maoming	Guangdong	China	Asia	
351	Shaoguan	Guangdong	China	Asia	
34	Shenzhen	Guangdong	China	Asia	
356	Laibin	Guangxi	China	Asia	
54	Liuzhou	Guangxi	China	Asia	
39	Rongan	Guangxi	China	Asia	
808	Zhongshan	Guangxi	China	Asia	
46	Qinzhou	Guangxi	China	Asia	
135	Lijiang	Yunnan	China	Asia	
94	Cang Mountain	Yunnan	China	Asia	
254	XishuangbannaTropical Botanical Garden	Yunnan	China	Asia	
77	Potatso National Park	Yunnan	China	Asia	
50	Tongzi	Guizhou	China	Asia	
38	Jiuzhaigou National Park	Sichuan	China	Asia	
132	Xinglong Tropical Botanical Garden	Hainan	China	Asia	
3	Hohhot	Inner Mongolia	China	Asia	
21	Changzhi	Shanxi	China	Asia	

Figure 1 Worldwide localization of samples, distribution of collected fungi in different climate zones, and some fungal morphology.

(A) Samples were collected worldwide, indicated by red dots. (B) Distribution of the collected fungi in different climates zones. The climate classification was based on the Köppen–Geiger Climate Classification. The codes of the climate are as the following: Aw: Tropical monsoon climate; Bsk: Cold semi-arid climate; Cfa: Humid subtropical climate; Cfb: Temperate oceanic climate; Cwa: Monsoon-influenced humid subtropical climate; Cwb: Subtropical highland climate or Monsoon-influenced temperate marine climate; Dfb: Humid continental climate; Dwa: Monsoon-influenced hot-summer humid continental climate; Dwb: Monsoon-influenced warm-summer humid continental climate. (C) Morphology of some isolated fungal colonies in the library.

Endophytic fungi from Chinese medicinal plants

Our fungal library includes many endophytic fungi. Since endophytic fungi produce many plant metabolites with medical functions, we collected plant samples from the three most extensive tropical botanic gardens in China, including South China, Hainan Xinglong, and Yunnan Tropical. There are many diverse plants in these gardens. We collected 27 well-known Chinese medicinal plants, such as Chinese black olive (Canarium pimela), Chinese croton (Excoecaria cochinchinensis), Lemon-scented gum (Eucalyptus citriodora Hook), Sweet osmanthus (Osmanthus fragrans), and Yellow cow wood (Cratoxylum cochinchinense) (Table 4). We separated different parts from each plant, e.g., leaf, bark, stem, fruits, and seeds, and sterilized their surfaces with 75% ethanol. Following grinding masses and culturing on Malt Extract Agar (MEA) plates, more than 50 endophytic fungi were isolated on MEA plates. The colors and morphology of these endophytic fungi and their corresponding plants look strikingly different (Fig. 2). The ITS of three fungi were PCR-amplified and sequenced (Accession # in GenBank are MT994711, MT994712, and MT594489). They were identified as Stephanonectria keithii (Fig. 2C), Aspergillus sp. (Fig. 2E), and Tolypocladium album (Fig. 2Z), respectively.

Table 3 The species of randomly sampled fungi from the library.

ID	Species	Length	Coverage	Identity	Source	Sample location	
11-A5	Acremonium cellulolyticus	581	100%	100%	soil	Guangxi, China	
S3/2	Albifimbria verrucaria	582	98%	100%	soil	Guangdong, China	
116-E12	Arthropsis hispanica	455	100%	100%	soil	Florida, USA	
4-H9	Ascomycota sp.	586	100%	100%	plant	Guangdong, China	
64-A1	Aspergillus flavus	597	100%	100%	plant	Oklahoma, USA	
79-B7	Aspergillus sp.	574	100%	100%	soil	Guangdong, China	
49-G11	Aspergillus sydowii	569	100%	100%	air	Guangdong, China	
58-C1	Aspergillus sp.	621	92%	77%	soil	Guangdong, China	
3-D7	Cylindrocladium sp.	575	100%	99%	soil	Guangdong, China	
24-C5	Debaryomyces subglobosus	634	100%	100%	soil	Guangdong, China	
116-H3	Epicoccum sorghinum	589	100%	100%	plant	Guangxi, China	
88-E10	Fusarium kyushuense	540	95%	100%	soil	Guangdong, China	
126-G10	Fusarium solani	573	100%	100%	soil	Guangxi, China	
117-B9	Fusarium solani	576	100%	100%	soil	Florida, USA	
45-F10	Fusarium solani	573	100%	100%	soil	Guangdong, China	
3-F5	Fusarium verticillioides	562	100%	100%	soil	Guangxi, China	
18-F5	Trichoderma sp.	619	99%	92%	plant	Hainan, China	
66-B4	Metarhizium sp.	617	100%	99%	soil	Yunnan, China	
17-A1	Mycosphaerella sp.	573	100%	100%	soil	Alaska, USA	
HW	Neurospora sp.	586	100%	99%	air	Guangdong, China	
74-F11	Penicillium rolfsii	591	99%	100%	soil	Guangdong, China	
99-H4	Penicillium sclerotiorum	586	100%	100%	plant	Guangxi, China	
78-D10	Penicillium sclerotiorum	583	100%	100%	soil	Guangdong, China	
81-D8	Penicillium sp.	575	100%	99%	soil	Guangdong, China	
107-H3	Penicillium soppii	470	100%	100%	soil	Alaska, USA	
125-B10	Penicillium sp.	589	100%	100%	soil	Guangdong, China	
114-D12	Penicillium sp.	584	100%	100%	plant	Yangon, Myanmer	
3-G10	Talaromyces stipitatus	582	100%	100%	soil	Guangxi, China	
37-A6	Trichoderma sp.	612	83%	86%	plant	Guangdong, China	
95-C6	Trichoderma sp.	571	100%	99%	soil	Guangxi, China	
45-G11	Trichoderma atroviride	629	100%	100%	soil	Guangdong, China	
CL	Trichoderma sp.	612	100%	99%	soil	Maasi Marla, Kenya	
16-D5	Trichoderma harzianum	625	100%	100%	soil	Guangdong, China	
100-D3	Trichoderma sp.	554	100%	99%	soil	Guangdong, China	
104-D12	Trichoderma harzianum	623	100%	100%	soil	Guangxi, China	
98-C8	Trichoderma sp.	610	100%	99%	soil	Guangxi, China	
80-G9	Trichoderma sp.	622	100%	99%	soil	Guangdong, China	
114-G8	Trichoderma sp. 	617	100%	99%	soil	Guangdong, China	
117-H9	Trichoderma harzianum	624	100%	100%	soil	Florida, USA	
121-G6	Metarhizium carneum	172	100%	100%	plant	Hainan, China	

Table 4 Some fungal strains isolated from Chinese medicinal plants.

ID	Sample collection location	Medical plant/tissue	Family	Species	
12A7	SCBG, Guangzhou, China	Chinese black olive/stem	Burseraceae	Canarium pimela	
30F12	SCBG, Guanzhou, China	Devil’s Trumpet/leaf	Solanaceae	Datura metel	
32F1	SCBG, Guanzhou, China	Japanese lantern	Malvaceae	Hibiscus schizopetalus	
38H8	SCBG, Guangzhou, China	Sweet osmanthus/leaf	Oleaceae	Osmanthus fragrans (Thunb)	
42B7	SCBG, Guangzhou, China	Lemon-scented gum/bark	Myrtaceae	Eucalyptus citriodora Hook.f	
44G1	SCBG, Guangzhou, China	Sausage Tree/leaf	Bignoniaceae	Kigelia africana (Lam.) Benth.	
44G9	SCBG, Guangzhou, China	Chinese croton/leaf	Euphorbiaceae	Excoecaria cochinchinensis	
71G5	SCBG, Guangzhou, China	Maytenus hookeri Loes/leaf	Celastraceae	Maytenus hookeri Loes.	
71G11	SCBG, Guangzhou, China	Siris tree/leaf	Fabaceae	Albizia lebbeck	
74E7	SCBG, Guangzhou, China	Yellow cow wood/leaf	Hypericaceae	Cratoxylum cochinchinense	
74E10	SCBG, Guangzhou, China	Dracontomelon duperreanum/leaf	Anacardiaceae	Dracontomelon duperreanum	
35H11	SCBG, Guangzhou, China	Harland boxwood/leaf	Buxaceae	Buxus microphylla	
9D1	HXTBG, Haikou, China	Sisal/leaf	Asparagaceae	Agave sisalana	
9H3	SCBG, Guangzhou, China	Candlenut/leaf	Euphorbiaceae	Aleurites moluccanus	
9H7	SCBG, Guangzhou, China	Kadsura vine/leaf	Schisandraceae	Kadsura japonica	
21B3	SCBG, Guangzhou, China	Castor oil plant/fruit	Euphorbiaceae	Ricinus communis	
29A7	SCBG, Guangzhou, China	Soft bollygum/leaf	Lauraceae	Litsea glutinosa	
37E10	SCBG, Guangzhou, China	Sweet osmanthus/leaf	Oleaceae	Osmanthus fragrans	
79A7	HXTBG, Haikou, China	Thyme/leaf	Lamiaceae	Thymus mongolicus Ronn	
79A11	Taishan, China	Sterculia nobilis/leaf	Malvaceae	Sterculia nobilis Smith	
79F5	SCBG, Guangzhou, China	Milk tree/leaf	Moraceae	Ficus hispida L. f.	
82C1	HXTBG, Haikou, China	Sessileflower/leaf	Araliaceae	Acanthopanax gracilistylus	
72E3	SCBG, Guangzhou, China	Trumpet tree/leaf	Urticaceae	Cecropia peltata	
81F7	SCBG, Guangzhou, China	Callicarpa brevipes/fruit	Lamiaceae	Callicarpa brevipes	
78A11	SCBG, Guangzhou, China	bollygum /leaf	Lauraceae	Litsea glutinosa	
78A10	SCBG, Guangzhou, China	African oil palm/leaf	Arecaceae	Elaeis guineensis	
126H5	HXTBG, Haikou, China	Giant crepe-myrtle/bark	Lythraceae	Lagerstroemia speciosa	
Notes.

SCBG South China Botanical Garden

HXTBG Hainan Xinglong Tropical Botanical Garden

Construction of a fungal extract library

To generate a fungal metabolite library, we used the cereal-based medium to produce the secondary metabolites as reported (Niu et al., 2015). Each fungus was cultured in a test tube with six small pieces of cereals. After culturing for one month, we used ethyl acetate to extract the secondary metabolites. We obtained 9,053 crude extracts in total, each of which corresponds to a specific fungal isolate. As anticipated, different fungal strains produced different amounts of secondary metabolites from 1 mg to 20 mg per gram culture, and have various physical features such as stickiness, odors, and solubility. More than 90% of the extracts have colors, including green, orange, red, yellow, purple, and others. The crude extracts were dissolved in DMSO to generate 2 mg/mL solution. For future reference, we named this library “Global Fungal Extract Library” or GFEL in brief.

Screen the fungal extract library against P. falciparum transmission to mosquitoes

Malaria remains a devastating disease, and Anopheles midgut protein fibrinogen-related protein 1 (FREP1) mediates Plasmodium transmission (Li et al. 2013; Zhang et al., 2015; Niu et al., 2017). FREP1 mediates Plasmodium invasion in mosquitoes by binding to P. falciparum gametocytes or ookinetes (Zhang et al., 2015).

To examine the usability of the newly constructed fungal library, we screened 460 fungal extracts obtained in later 2016 and early 2017 for their inhibition activity against FREP1-P. falciparum interaction and found 4 extracts that prevented FREP1 from binding to P. falciparum lysates by over 90%. The fungal colonies of these four fungal candidates on MEA plates showed different colors and shapes (Figs. 3A–3D). Then, we determined the activities of the four candidates against P. falciparum to An. gambiae in vivo using the standard membrane feeding assays (SMFA). Results show that the fungal extracts of 37C6 and 22E8 could block malaria transmission at a concentration of 100 µg/mL, and 100D3 and 45F10 did not (Fig. 3E, File S2). The oocyst numbers of 37C6 or 22E8 extract-treated mosquitoes were nearly zero, while the oocyst numbers in the 100D3 or 45F10 extract-treated mosquitoes were not significantly different from that of the DMSO control (Fig. 3E). After further dilution of the two positive candidate fungal extracts (37C6 and 22E8) to 20 µg/mL, we found that the extract of 22E8 still significantly inhibited the activity in P. falciparum transmission to mosquitoes (Fig. 3F, File S2). We sequenced the ITS sequences of the four candidate fungi and their accession number are MT594486–MT594488 and MT613342 at GenBank at https://www.ncbi.nlm.nih.gov/genbank/sequenceids/. According to ITS sequences of 22E8 (Acc #: MT613342) and 37C6 (Acc#: MT594487), the fungal species of 22E8 and 37C6 were Penicillium thomii and Penicillium pancosmium, respectively (Table 5). Notably, this is the first report about P. thomii and P. pancosmium that produce secondary metabolites with antimalarial activities. An independent project in our lab identified Asperaculane B as an active compound from this GFEL that inhibited malaria transmission to mosquitoes (Niu et al., 2020).

Figure 2 The fungal isolates and the corresponding Chinese medicinal host plants.

The images in A–I, S–AA and JJ–SS were the fugal isolates grown on MEA plates, and the photos below in J–R, BB–II and TT–BBB are the corresponding host plants where they were isolated.

Figure 3 The transmission-blocking activity of the extracts candidates by screening the fungal library with the in vitro FREP1-parasite interaction-based ELISA assays.

(A–D) The morphology of fungal isolates 37C6, 100D3, 22E8T, and 45F10 on the MEA agar plate. (E) The final concentrations of the fungal extracts were 100 µg/mL and the results show that the fungal extracts (37C6 and 22E8) significantly reduced the oocyst number compared with the DMSO control while the oocyst number of the other two (100D3 and 45F10) was not significantly different with the DMSO control, respectively. (F) Further, the fungal extract of 22E8 continued to show a significant reduction of the oocyst number in midgut while the 37C6 fungal extract did not have significant effects on P. falciparum infection in mosquitoes when the concentration of the fungal extracts was decreased to 20 µg/mL. N: the number of mosquitoes for each treatment; mean: the average number of oocysts per midgut; PR: infection prevalence in mosquitoes. p: the p-value was calculated by the Mann-Whitney-Wilcoxon test. The experiments were repeated three times.

Examine the usability of fungal extract library in finding antibiotic leads

Antibiotic-resistant bacteria threaten public health (Todd, 2017). We screened potential antibiotics from this newly established fungal library against antibiotic-resistant bacterial pathogens. We randomly picked 574 fungal extracts and examined their activity in inhibiting Gram-positive (methicillin-resistant S. aureus MRSA) and Gram-negative (S. flexneri) bacterial pathogens. These fungi were isolated from samples collected in late 2017. Among them, 47 inhibited the growth of S. aureus MRSA (hit rate of 10.8%), and one inhibited the growth of S. flexneri (hit rate of 0.17%, Table 6). The hit rate of antibiotics against Gram-positive bacteria was higher than that of the Gram-negative bacteria, which is consistent with the well-known challenges in antibiotic discovery against gram-negative pathogens. Gram-negative pathogens have unique outer membrane and efflux pumps (Fair & Tor, 2014).

We also examined the effect of 288 fungal extracts on non-methicillin-resistant S. aureus. The results showed that 22 prevented the growth of non-methicillin-resistant S. aureus (hit rate of 7.6%). Besides, we analyzed another two featured bacteria, Mycobacterium smegmatis, which is gram-positive and has acid-fast dye staining cell wall, and E. coli-AS17tolc, which has gram-negative cell wall, but more permeable than the wild type E. coli. We obtained eight candidates against Mycobacterium smegmatis and three candidates against E. coli-AS17tolc. The results showed 10.8% and 1% hit rates to Mycobacterium smegmatis and E. coli-AS17tolc, respectively (Table 6 and File S3). The results show that the secondary metabolites produced by different types of fungi in the GFEL contained biologically active compounds against drug-resistant and pathogenic strains of bacteria.

Table 5 The fungal strains producing metabolites that inhibited malaria transmission.

			aTransmission blocking activity (%)	
ID	Fungal species	Access No	100 µg/mL	20 µg/mL	
22E8	Penicillium thomii	MT613342	99.4	44.9	
100D3	Trichoderma harzianum	MT594486	NS	NS	
37C6	Penicillium pancosmium	MT594487	99.3	NS	
45F10	Fusarium solani	MT594488	NS	NS	
Notes.

a The transmission-blocking activity was calculated with the equation: (the mean oocyst number of the control group—the mean oocyst number of the treatment group)/the mean oocyst number of the control X100%.

NS Not significant

Table 6 Antibiotic screening results show the positives and hit rates to various bacteria.

	Mycobacterium smegmatis	Staphylococcus aureus (WT)	Staphylococcus aureus (MRSA)	Shigella flexneri	E.coli (AS17tolc)	
Total extracts screened	574	288	574	574	288	
Number of positive hits	62	22	47	1	3	
Hit rate (%)	10.8	7.6	8.2	0.17	1	

Examine the usability of fungal extract library in finding anti-chronic myeloid leukemia candidates

Finally, we studied the possibility of obtaining drug candidate leads against chronic myelocytic leukemia (CML), a malignant tumor of the blood system (Kaleem et al., 2015). A small subset of endophytic fungi from Chinese medicinal plants was used for this purpose. Fifty extracts were examined for their inhibition against the human immortalized myelogenous leukemia cell line K562 using the MTT method. The final concentrations of crude extracts were 20 µg/mL. The results from triplicates showed that 4 extracts (#17, #22, #29, #35) significantly inhibited the proliferation of K562 cells (p < 0.05, Fig. 4, File S4). The hit rate of the plant-fungal metabolite library was 8%. Notably, the survivorship of K562 cells with extract #29 was about 24.8% compared with the control, e.g., the inhibition of extract #29 on K562 proliferation was as high as 75.2%. The specimen of #29 was from the fungus (Fig. 2Z) isolated from the plant Litsea glutinosa (Fig. 2HH) collected from the Medical Botanical Garden of South China Botanical Garden in Guangzhou. Litsea glutinosa has been used to treat diarrhea, traumatic injuries, mumps, and rheumatism. It contains abundant flavonoids, terpenes, and alkaloids, and has good antibacterial activity, immunomodulatory effect, and anti-tumor effect. We PCR-amplified the ITS region of the fungal candidate (#29) and sequenced the region. Based on the ITS region sequence (Accession # at GenBank: MT594489), the candidate fungus was Tolypocladium album. Notably, a tetrameric acid from Tolypocladium album has been reported to inhibit the tumor’s growth (Fukuda et al., 2015). Further studies will be conducted to isolate and identify the bioactive compounds from this fungus.

Figure 4 Anti-chronic myeloid leukemia screening results.

About 50 extracts of endophytic fungi isolated from the Chinese medicinal plant-fungal metabolite library were examined against K562 cells using the MTT method. The results show that 4 extracts (#17, #22, #29, #35) significantly inhibited the growth of K562 cells (p < 0.05). Notably, the survivorship of K562 cells with extract #29 was about 24.3% compared with the control, or the inhibition of extract #29 on K562 was 75.7%. DMSO was applied as the control. The data shows the means and standard deviations of triplicates.

Conclusions

We established a comprehensive fungal library that is diverse and useful for the scientific communities. We also demonstrated the usability of GFEL and identified a set of fungal strains that produce secondary metabolites to inhibit Plasmodium falciparum’s transmission, chronic pneumonia development, and bacteria proliferation. Further studies will identify the active compounds for drug development.

Supplemental Information

Supplemental Information 1 Phylogenetic tree of the 40 fungi

The phylogenetic tree was constructed based on their ITS sequences. Three large genus groups predominantly presented in soils and associated with plants were labeled.

Click here for additional data file.

Supplemental Information 2 Numeric raw data show the number of oocysts in Fig 3B and 3C

Click here for additional data file.

Supplemental Information 3 Raw data show candidate fungal extracts against bacterial proliferation

Click here for additional data file.

Supplemental Information 4 Numeric raw data of Fig 4 show fungal extracts inhibiting K562 proliferation

Click here for additional data file.

Supplemental Information 5 Fungal ITS sequences

All sequences but the accession number MT613342, stain name 22E8 are available at https://www.ncbi.nlm.nih.gov/nuccore/a ccess#. The ITS sequences for accession number MT613342 will be available in next GenBank release.

Click here for additional data file.

We thank numerous residents and colleagues for collecting samples from various places and mailing the samples to us. We also appreciated many undergraduate students and technicians in culturing and isolating fungi.

Additional Information and Declarations

Competing Interests

Author Contributions

Field Study Permissions

DNA Deposition

Data Availability

The authors declare there are no competing interests.

Guodong Niu and Jun Li conceived and designed the experiments, performed the experiments, analyzed the data, prepared figures and/or tables, authored or reviewed drafts of the paper, and approved the final draft.

Thirunavukkarasu Annamalai, Xiaohong Wang and Guomin Niu performed the experiments, analyzed the data, prepared figures and/or tables, and approved the final draft.

Sheng Li and Stephen Munga performed the experiments, authored or reviewed drafts of the paper, and approved the final draft.

Yuk-Ching Tse-Dinh conceived and designed the experiments, analyzed the data, authored or reviewed drafts of the paper, and approved the final draft.

The following information was supplied relating to field study approvals (i.e., approving body and any reference numbers):

Field collection was approved by the United States Department of Agriculture on 9/12/2018 (permit number P526-180522-001). This USDA permit allowed us to receive global samples legally.

The following information was supplied regarding the deposition of DNA sequences:

All fungal ITS sequences are available at GenBank: MT584204, MT594355–MT594393, MT594486, MT594487, MT594488, MT994711–MT994712, and MT613342.

The following information was supplied regarding data availability:

The ITS sequences, phylogenetic tree of 40 fungal isolates from Table 3, raw numeric measurements, and the inhibition results of fungal extracts against different bacterial strains are available in the Supplemental File.

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
