# Peer review of "A diverse global fungal library for drug discovery"

_PeerJ, doi:10.7717/peerj.10392_

## Round 0.1 · original submission · Major Revisions

Please address the reviewers' questions and concerns. In particular, Reviewer 1 brings up a number of questions regarding the experimental methods that should be responded to.

Reviewer 1 ·

Basic reporting

With regard to the manuscript titled "A diverse global fungal library for drug discovery," I believe the work can be published. Building a fungal library with such a wide diversity and at a global level is very useful for high throughput screening of fungal secondary metabolites for bioactive compounds.

I am concerned about the taxonomic names of the fungi, please see my comment below.

I think the reference citations can be improved. For some, reason I felt the authors have few references cited.

Figure 2. The plates are all streaked like bacteria, but in Mycology that is not usually how the fungal transfers are performed - just a suggestion

Figure 4b. Really cannot see anything clearly. Also, check the spelling of DMSO

Experimental design

No Comment

Validity of the findings

See below

Additional comments

Chemistry and Pharmacology testing

Comment #1: Citing references is needed for the information stated at lines 46-47, lines 47-48, and line 52.

Comment #2: Time units such as seconds, minutes, and hours can be abbreviated as sec, min, and h, respectively.

Comment #3: The picture used in figure 2, it is better to add a label on the left side of rows 1, 3, and 5 as fungal isolates and rows 2, 4, and 6 as host plants.

Comment #4: In figure 4, authors need to specify whether the anti-chronic myeloid leukemia screening was performed in triplicate, and what the error bars actually represent?

Comment #5: It is nice to see that crude extracts were obtained for the total 9,053 fungal strains. It would be good to study the chemical diversity of these extract samples and develop a dereplication method for the purpose of discovering novel bioactive leads.

Comment #6: Out of the 9, 053 crude fungal extracts, some were randomly selected for antimalarial, antimicrobial, and cytotoxic activities. The authors need to identify the method followed to randomly select these samples for screening. Were these samples representative for all different locations and climate conditions. Were there any extracts that were tested against more than one screening assay?

Comment # 7: The authors might in the future make a fraction library from their extracts and then perform bioassays. Each extract can undergo flash chromatography to make the fractions, which can then be tested against specific bioassays. Thus when activity is found it would help zoom into the compound peak that is responsible for the activity.


Mycology

Comment #1 Reference for ITS primers is not given.

Comment #2 The identification to species level with the ITS region might not necessarily be accurate. Especially for genera like Penicillium and Fusarium. Thus, please only use the genus name for identification. All Table 1 species names might not be accurate unless they are identical to the type sequence for ITS in GenBank. Thus the authors need to thoroughly check their identification to species-level with the ITS region just via BLAST search in NCBI GenBank with no consultation with type databases

Comment #3 Did the authors follow the Nagoya protocol for collections in Africa? I don't see any collaborator from an African University/institution?

Reviewer 2 ·

Basic reporting

Review of: A diverse global fungal library for drug discovery

The manuscript entitled “A diverse global fungal library for drug discovery” describes the construction of a fungal extract library containing extracts from 9053 fungal isolates obtained from approximately 5000 (soil and plant samples harvested in 36 regions in Africa, Asia, and North America. The biological activity of randomly selected extracts was evaluated in several assays including P. falciparum transmission to mosquitoes, antibacterial, and anti-chronic myeloid leukemia.

Generally, this manuscript is well written, however, some references are missing in the introduction section. Additionally, reference 9 is not appropriate for the text, it does talk about actinomycetes instead of ascomycetes.
If well, the isolation of 9053 fungal microorganisms is a titanic task, the information provided about the diversity in the chemistry and taxonomy of the isolates is not well presented in the submitted version; a phylogenetic tree must be ideal to represent the taxonomic distribution of the isolates. More importantly, the manuscript does not present any evidence of the chemistry of the isolates, possibly, an untargeted metabolomics study will provide such information. Ideally, the isolation and structure elucidation of some bioactive compounds is recommended.

Taking into account the above concerns and the suggestions marked in the attached pdf file, the manuscript must be accepted after major revisions.

Experimental design

The experimental design is appropriate.

Validity of the findings

Personally, I do believe the manuscript does not provide any new information.

Additional comments

Review of: A diverse global fungal library for drug discovery

The manuscript entitled “A diverse global fungal library for drug discovery” describes the construction of a fungal extract library containing extracts from 9053 fungal isolates obtained from approximately 5000 (soil and plant samples harvested in 36 regions in Africa, Asia, and North America. The biological activity of randomly selected extracts was evaluated in several assays including P. falciparum transmission to mosquitoes, antibacterial, and anti-chronic myeloid leukemia.

Generally, this manuscript is well written, however, some references are missing in the introduction section. Additionally, reference 9 is not appropriate for the text, it does talk about actinomycetes instead of ascomycetes.
If well, the isolation of 9053 fungal microorganisms is a titanic task, the information provided about the diversity in the chemistry and taxonomy of the isolates is not well presented in the submitted version; a phylogenetic tree must be ideal to represent the taxonomic distribution of the isolates. More importantly, the manuscript does not present any evidence of the chemistry of the isolates, possibly, an untargeted metabolomics study will provide such information. Ideally, the isolation and structure elucidation of some bioactive compounds is recommended.

Taking into account the above concerns and the suggestions marked in the attached pdf file, the manuscript must be accepted after major revisions.

Annotated reviews are not available for download in order to protect the identity of reviewers who chose to remain anonymous.

---

## Round 0.2 · Minor Revisions

Thank you for addressing the reviewer comments. There are just a few minor comments that could be addressed. In particular, please do provide more details on the construction of the phylogenetic tree. The description of parameters used (even if they are defaults) would be particularly valuable if someone is trying to generate this tree.

Reviewer 1 ·

Basic reporting

N/A

Experimental design

N/A

Validity of the findings

N/A

Additional comments

The authors have addressed most of my comments, but some very minor issues remain.

1. "sp." should not be italicized throughout.
2. Please use the original citations for the ITS primers utilized.
3. The authors have oversimplified the making of the phylogenetic tree. Please elaborate a bit.

---

## Round 0.3 · accepted · Accept

Thank you for addressing the reviewer's concerns and congratulations again!